# Effects of Metal and Fluoride Powders Deposition on Hot-Cracking Susceptibility of 316L Stainless Steel in TIG Welding

**Kamel Touileb** [1,*], **Abousoufiane Ouis** [1], **Abdeljlil Chihaoui Hedhibi** [1,2], **Albaijan Ibrahim** [1] **and Hany S. Abdo** [3,4]

1    Department of Mechanical Engineering, College of Engineering in Al-Kharj,
     Prince Sattam Bin Abdulaziz University, P.O. Box 655, Al-Kharj 16273, Saudi Arabia;
     a.ouis@psau.edu.sa (A.O.); a.hedhibi@psau.edu.sa (A.C.H.); i.albaijan@psau.edu.sa (A.I.)
2    Department of Mechanical Engineering, National Engineering School of Tunis (ENIT), El-Manar University,
     P.O. Box 37, Tunis 1002, Tunisia
3    Center of Excellence for Research in Engineering Materials (CEREM), King Saud University, P.O. Box 800,
     Al-Riyadh 11421, Saudi Arabia; habdo@ksu.edu.sa
4    Mechanical Design and Materials Department, Faculty of Energy Engineering, Aswan University,
     Aswan 81521, Egypt
*    Correspondence: k.touileb@psau.edu.sa

**Abstract:** This study aims to investigate the effects on the hot cracking susceptibility of fluoride powders such as $CaF_2$, NaF, LiF, and metal powders such as Mn, Ti, Nb and mixed Ti-Nb deposited on the 316L stainless steel during the TIG (Tungsten Inert Gas) welding process. A self-restraint hot cracking bench test using specimens of trapezoidal shape and 3 mm of thickness was selected. The obtained results of the weldability with the different powders were compared with those obtained with the conventional TIG parent-metal weld. The susceptibility to hot cracking was evaluated by the length of the crack and by the critical width at the end of the crack propagation. The formed cracks were first revealed by the liquid penetrant test, and then the surfaces of cracks were observed and analyzed by SEM-EDS-XRD tools. Among the powders tested, single Nb powder and the mixed flux of 80% Nb + 20% Ti exhibited the lowest crack length. The crack propagation ended at 22 mm of length and 30.8 mm of width. The analyses of the fracture surfaces of cracks revealed the presence of Niobium carbide ($Nb_2C$), titanium, chromium, niobium oxide ($TiO_{0.6}Cr_{0.2}Nb_{0.0202}$) complex compounds and cementite ($Fe_3C$) at the interdendritic zones.

**Keywords:** hot cracking; trapezoidal specimen; crack length; critical width; dendrites





## 1. Introduction

   Austenitic stainless steels are widely used in power systems, especially in the nuclear power industry where reliability is a primary requirement. The properties sought for these materials involve excellent resistance to corrosion, absence of ductile to brittle transition phenomenon and good properties at high temperatures, including resistance to oxidation.

   Stainless steel welds of the AISI/AWS 300 series can be classified into three types: A, B and C, according to their general microstructure and the morphology of the delta ferrite [1,2]. Type A welds, which have a low $Cr_{eq}/Ni_{eq}$ ratio (<1.48), solidify primarily as austenite phase, and then the delta ferrite, if any, forms from the remaining melt between the austenite cells. However, type B welds with a $Cr_{eq}/Ni_{eq}$ ratio range from 1.48 up to 1.95 solidify in a ferritic-austenitic phase, whereas type C welds, having a high $Cr_{eq}/Ni_{eq}$ ratio (>1.95), solidify as single-phase ferrite.

   Hot cracking is a major problem encountered when welding austenitic stainless steels. Factors causing the cracking of the weld metal during the solidification can be ascribed to metallurgical or mechanical phenomena. The metallurgical factors are related to the

conditions of the metal solidification, presence of low-melting eutectic films, etc. The mechanical factors are linked to the conditions of stress–strain raised in the weld metal during the solidification stage. Thus, mechanical tensile strains are generated due to weld metal solidification shrinkage, thermal contraction of the parent metal and external restraint of the welded structure [3].

It is known that hot cracking occurs when a low melting eutectic phase is present during the last solidification period of the molten metal. It is also well known that sulfur and phosphorus can combine with iron to form low melting point compounds such as FeS (1190 °C or 2174 °F) and $Fe_3P$ (1166 °C or 2131 °F), which in turn could form low melting eutectic compounds such as FeS-Fe (988 °C or 1810 °F) and $Fe_3P$-Fe (1050 °C or 1922 °F). Other low melting eutectics exist, such as $Ni_3S_2$-Ni (637 °C or 1179 °F) and $Ni_3P$-Ni (875 °C or 1607 °F). These compounds, which increase hot cracking susceptibility, are considered to form when sulfur and phosphorus segregate at the austenitic grain boundaries during solidification [4].

Hot cracks in welds are classified into three specific types, namely solidification cracking, liquation cracking and ductility dip cracking. Solidification cracking happens in the interdendritic regions in weld metal, whereas liquation and ductility dip cracking occur intergranularly in the heat-affected zone (HAZ). Fully austenitic weld metals are highly susceptible to solidification cracking, whereas primary delta ferritic solidification mode results in greatly increased cracking resistance [5].

The literature review shows several test methods for assessing hot cracking resistance. The used methods can be classified into external restraint and self-restraint tests. Examples of external restraint tests are the PVR test [6], Varestraint test [7] and Transvarestraint test [8]. Self-restraint tests mainly comprise the Circular patch test [9], Houldcroft test [10], Fan-shaped test [11] and Free-edge test [12]. However, several studies utilize the laser welding process to assess the hot cracking susceptibility. For instance, Wenbin et al. carried out solidification hot cracking investigation on SUS310 stainless steel with trapezoidal-shaped specimens [13]. They deduced that with an increase in the welding speed from 1.0 to 2.0 m/min during the laser welding process, the solidification cracking susceptibility remarkably decreased. Though the laser process is a high-speed welding with less weld distortion, it has, unfortunately, limited uses in industries due to the high cost of the welding machine. Moreover, Abu-Aesh et al. proposed a mechanism to analyze the hot cracking formation during the welding of fan-shaped test specimens using the Pulsed-Current Gas Tungsten Arc Welding process [14]. They confirmed that the thermal stresses in conjunction with the applied strains on the weld joint played a great role in the crack expansion.

In our study, the TIG process has been used which is widespread in overall industries. In order to evaluate the solidification cracking susceptibility, among the previously cited tests, we selected the JWRI (Joining and Welding Research Institute) tests inspired by the work of Houldcroft test, also known as fishbone test [15]. In this process, a self-restrained test was used because of its simplicity on the one hand and on the other hand, it was close to the real welding manufacturing situation simulating the industrial welding conditions. Thus, the widest edge of a trapezoid-shaped specimen was clamped, and the weld operation started from the narrow edge. A paste resulting from the mixture of the metal powders and methanol was deposited along the centerline of the specimen before welding. The fusion line was performed from the furthest part (narrow part) of the specimen which favors the appearance of a crack towards the widest part. The restraint will influence the strain fields around a moving weld pool resulting in a crack susceptibility. The thermal and shrinkage stresses generated during the welding interacted with the restraining forces to influence strain around the moving weld pool [16]. The hot cracking defect appeared as soon as the bead was started. The crack stopped when the width of the specimen reached a critical width.

The addition of external agents to the weld pool for hot cracking study can be performed through the filler wire such as the study carried out by Da silva et al. [17]. Varestraint and Transvarestraint tests were used to assess solidification cracking susceptibility

in welding of aluminum when external agent added to the weld pool was filler metal. They conclude that the GMAW technique with double pulsation proved to reduce cracking susceptibility during aluminum welding, in comparison with the conventional pulsed GMAW. The external agent can be conveyed to the weld pool through atomized sprayed powder. Recently, study performed by Sangwoo et al. [18] allowed us to compare the laser weldability between the base material and TiN-coated material. The effect of a TiN spray coating on aluminum to improve laser welding issues such as cracking susceptibility using the free-edge test. The rectangular plate shape as specimen has been used. They conclude that the effect of TiN coating on Al 6061-T6 contributed to reduce the high temperature cracking susceptibility.

The purpose of this work is the study of the effects of different metal powders deposited on the centerline of the trapezoidal specimen on the solidification cracking resistance. There are no comparative studies in the published literature devoted to adding external agents in powder form to reduce or avoid solidification cracking. This work aimed to propose a new route to treat cracks and avoid their appearance by using mono or compound materials before starting the welding process. This method opens a field for scientific research in innovative ways to select and study materials that contribute to reducing cracks. It is considered as a flexible method. Additive materials can be applied using a paintbrush. The solidification cracking susceptibility is assessed using the crack length and the critical width. The crack surfaces of the specimens with lowest crack lengths are investigated using SEM, EDS and XRD tools. The effects of the constituent elements in the powders on solidification cracking is discussed.

## 2. Materials and Methods

### 2.1. Material

The material used in this study was 316L austenitic whose chemical composition is shown in Table 1. There are a number of different methods for calculating the $Cr_{eq}$ and $Ni_{eq}$ values; however, for this work, the equations based on the WRC-1992 were used [19,20]. According to WRC-1992, $Cr_{eq}$ = 18.64 and $Ni_{eq}$ = 12, so the ratio $Cr_{eq}/Ni_{eq}$ = 1.55. This grade solidifies in a ferritic-austenitic Type B welds with a ratio $Cr_{eq}/Ni_{eq}$ ranging from 1.48 up to 1.95.

**Table 1.** Chemical composition of 316L stainless steel.

| Elements | C | Mn | Si | P | S | Cr | Ni | Mo | N | Cu | Fe |
|----------|-----|------|------|-------|--------|-------|-------|------|-------|------|---------|
| Weight % | 0.026 | 1.47 | 0.42 | 0.034 | 0.0016 | 16.60 | 10.08 | 2.14 | 0.044 | 0.50 | Balance |

The deposited powders on the centerline of the trapezoid specimen are listed in Table 2.

**Table 2.** Powders, melting and evaporation temperatures.

| Powders | Melting Temperature (°C) | Evaporation Temperature (°C) |
|---------|--------------------------|------------------------------|
| CaF2 | 1418 | 2533 |
| LiF | 848 | 1673 |
| NaF | 993 | 1700 |
| Mn | 1246 | 2061 |
| Nb | 2477 | 4744 |
| Ti | 1668 | 3287 |

### 2.2. Welding Procedure

The tests were carried out on 'trapezoidal' JWRI specimens prepared using a laser beam machine in the rolling direction of an austenitic stainless steel 316L sheet, the thickness

of which was 3 mm. Test specimens were shaped according to Houldcroft's test pattern and the dimensions were inspired by those used by Kerrouault [21] (see Figure 1a,b). The specimens were notched by a gap of 1 mm at 20 mm from the free edge of the specimen which allows the initiation of the crack propagation during the steady state. The past consisted of a mixture of powders and methanol at a ratio of 1:1 was deposited along the centerline of each specimen. Afterwards, a TIG welding fusion line was performed along this centerline. The arc started from tab plate (narrow part) as shown in Figure 1c. When the fusion reached the bridge (ligament), tab plate (tailpiece) fell, and a steady fusion continued on the test specimen. Then, the hot cracking defect appeared as soon as the bead was initiated. The crack stopped when the width of the specimen reached a critical width. The weld metal was strained in a direction transverse to the welding direction. To ensure the accuracy in well-established test procedure, three tests were carried out for each type of selected powder, including the specimen without deposit powder. The mean values of face side bead width and root bead width were calculated. The crack susceptibility was assessed based on the mean measurements of crack length and mean critical crack width. Spherical powder with average size in the range of 325 mesh was heated separately in the furnace at 180 °C for 1 h to eliminate humidity. The powder was mixed with acetone with (1:1 ratio) and made in the form of paste; a paintbrush was used to apply the mixture on a plain surface to be welded. The acetone was allowed to evaporate, resulting in only the powder being left on the surface of the metal. The mean coating density of the deposit was about 3–4 mg.cm$^{-2}$, and the paste layer thickness about 0.2 mm. Before the welding operation, the plates were cleaned with acetone.

The test set up is shown in Figure 2a. The specimens were cleaned with acetone. The powders were dried in the furnace for 1 h at 180 °C to eliminate the humidity. Then, a thin layer of the powder mixed with methanol was applied using a brush on the surface subjected to the welding, as shown in Figure 2b. The welds were executed using the motorized TIG welding machine. The electrode used had a diameter of 3.2 mm and the torch was mounted on a motorized carriage, as shown in Figure 2. Figure 2c shows the specimen after welding where the crack is located at the centerline of the coupon.

Welding parameters can have a significant impact on the cracking susceptibility [22]. At higher speeds, the weld center becomes dominated by equiaxed grains, which significantly improves the weldability [23]. Welding parameters were thoroughly chosen to ensure a full penetration weld, a width weld face close to the width weld back, a stable arc and a sound weld. The selected experimental welding parameters are presented in Table 3.

**Table 3.** Welding parameters.

| Parameter | Value |
| --- | --- |
| Welding current | 150 A |
| Voltage | 11 V |
| Efficiency | 75% |
| Welding speed | 28 cm/min |
| Heat provided | 265 J/mm |
| Shield gas face | Argon 8 L/min |
| Shield gas backside | Argon 6 L/min |
| Electrode diameter | 2.4 mm |
| Electrode type | Tungsten thoriated 2% |
| Electrode tip angle | 45° |
| Arc length | 2 mm |
| Welding mode | Negative direct current electrode |

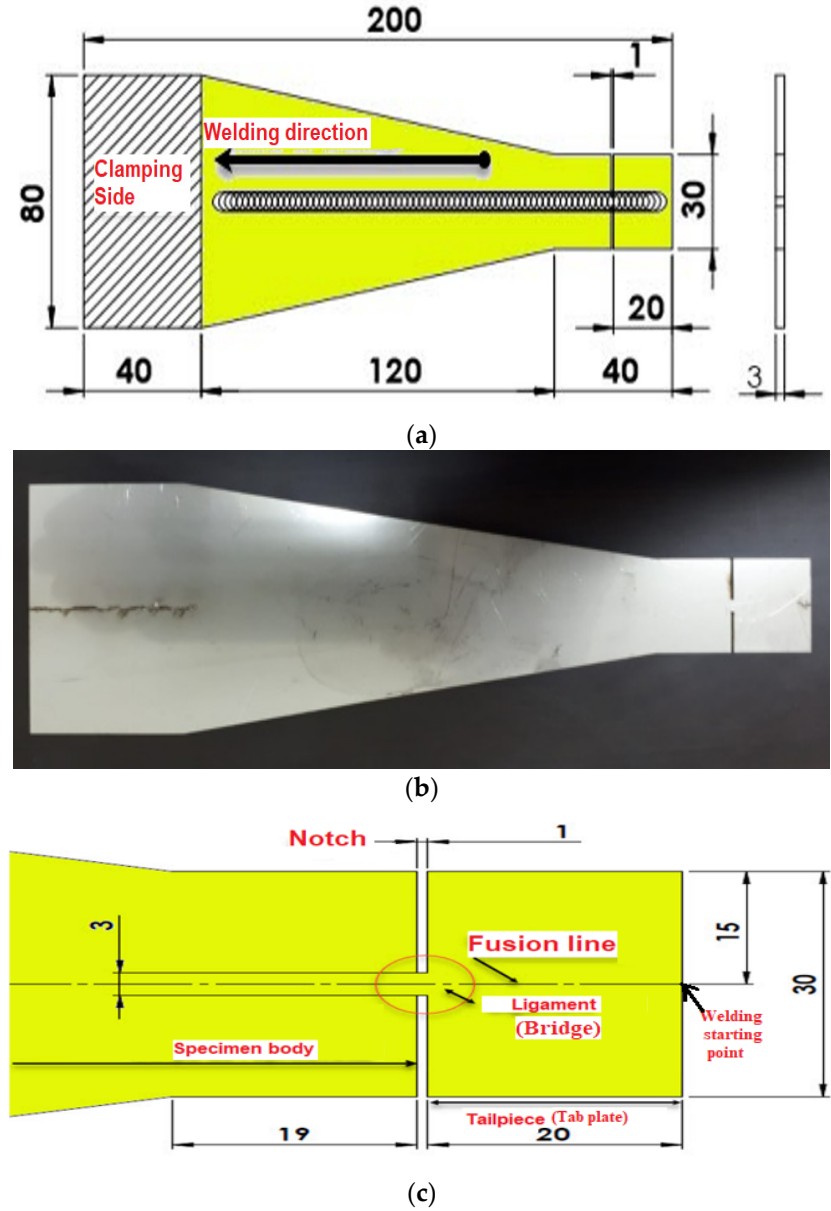

**Figure 1.** Trapezoidal specimen (**a**,**b**) and tailpiece (Tab plate) details (**c**). Dimensions in mm.

First, the crack in the weld shown in Figure 3b was revealed by the liquid penetrant nondestructive test using the different liquids shown in Figure 3a, and then the crack length was accurately measured.

After the welding process, the samples were cut far from the welding starting point to ensure that the arc welding was stabilized, as shown in Figure 2.

The 316L weld were etched using a glyceregia solution (15 mL HCl + 5 mL HNO$_3$ + 10 mL glycerol). Afterwards, the morphology of the welds was checked using an optical Motic Images plus version 2.0 software integrated with a microscope CAROLINA (CAROLINA, Burlington, NJ, USA). Microstructural characterization of the welded specimens was performed. The assessment of the fusion zone has been analyzed in the locations shown in Figure 4. Micrographs were taken on a JEOL JSM-7600F scanning electronic microscope (SEM) (JEOL, Tokyo, Japan). SEM with energy dispersive spectrometer (EDS) and X-ray diffraction were used for the weld analysis at the lateral solidification hot cracking facies, as shown in Figure 4.

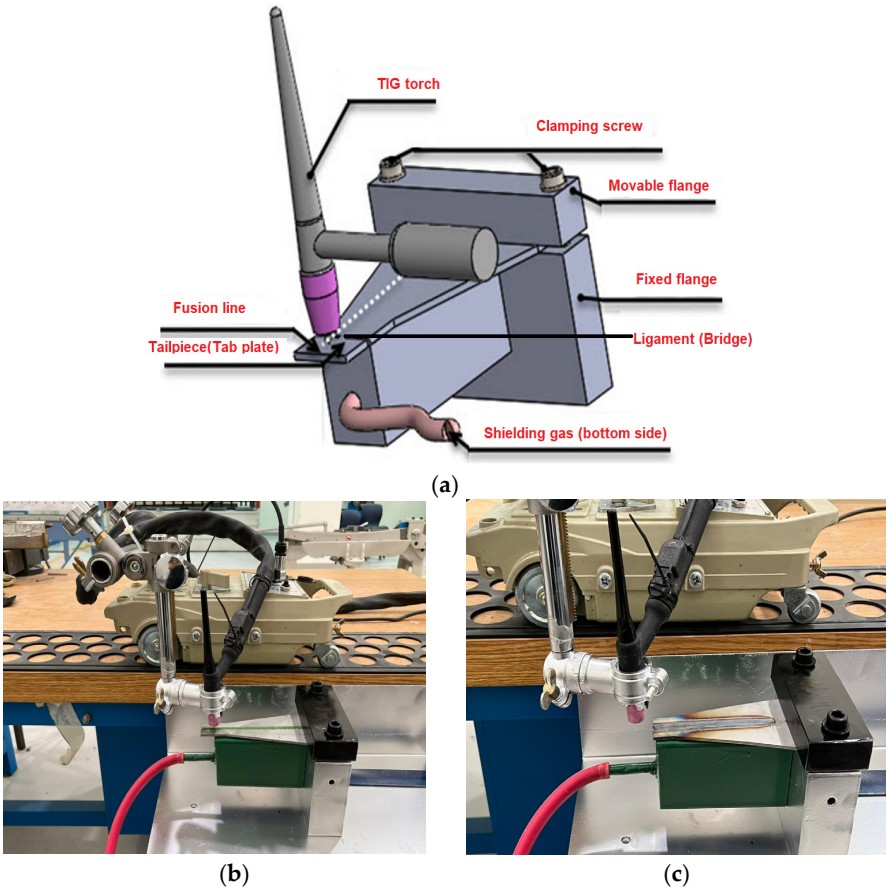

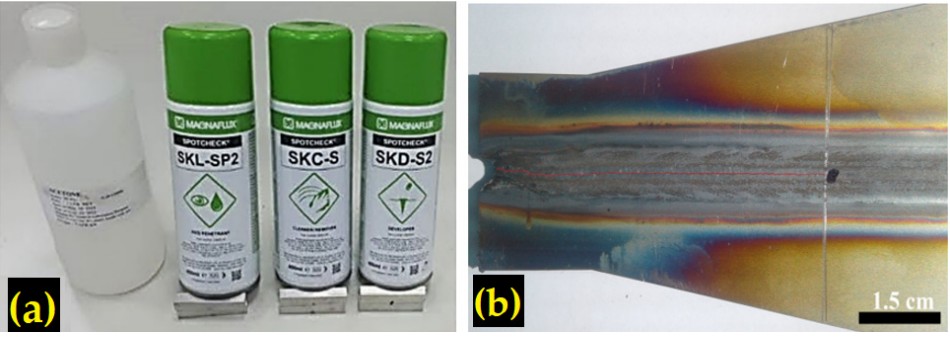

**Figure 2.** Test setup sketch (**a**), specimen before welding (**b**) and specimen after welding (**c**).

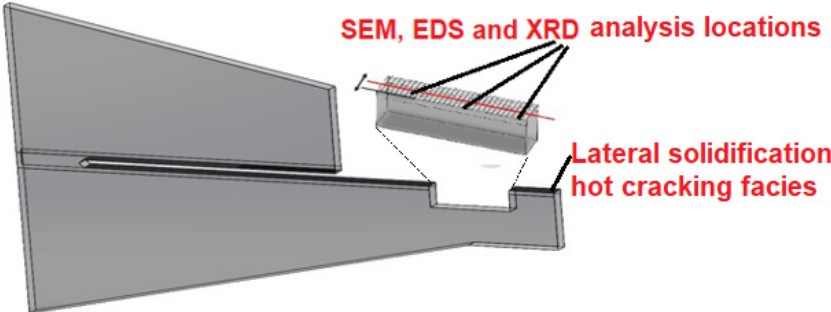

**Figure 3.** Liquid penetrant kit used after welding (**a**) on the weld line (**b**).

**Figure 4.** Microstructural characterization analysis locations.

## 3. Results and Discussions

### 3.1. Crack Length and Critical Width

Crack lengths and critical widths of the TIG welding lines performed on trapezoidal specimens have been compared and discussed for the following treatments: specimens without any coating powder and specimens coated with different powders ($CaF_2$, LiF, NaF, Ti, Mn, Nb and mixed Nb-Ti). Crack length and critical width were considered when the crack stopped its propagation. The results are listed in Table 4. The results obtained allowed us to find the best coating powder leading to the best resistance to hot cracking. Consequently, the specimen with the highest hot cracking resistance was preferentially chosen for SEM, EDS and XRD analyses.

**Table 4.** Weld morphology, crack length, critical width and standard deviation.

| TIG Weld | Number of Tests | Mean Face Width Weld Bead (mm) | Mean Back Width Weld Bead (mm) | Mean Crack Length Face Side (mm) | Standard Deviation σ For Crack Length Face Side | Mean Crack Length Back Side (mm) | Mean Critical Crack Width Face Side (mm) | Standard Deviation σ For Critical Crack Width Face Side | Mean Critical Crack Width Back Side (mm) |
|---|---|---|---|---|---|---|---|---|---|
| Without powder | 3 | 8 | 5 | 74 | 0.88 | 76 | 52.5 | 0.78 | 53.3 |
| $CaF_2$ | 3 | 6 | 4 | 49 | 0.79 | 50 | 42.3 | 0.72 | 43.7 |
| LiF | 3 | 6 | 5 | 48 | 0.96 | 49 | 41.7 | 1.03 | 42.1 |
| NaF | 3 | 6 | 4 | 40 | 1.00 | 44 | 38.3 | 1.08 | 40.0 |
| Ti | 3 | 6 | 5 | 56 | 1.07 | 56 | 45.0 | 1.17 | 1.17 |
| Mn | 3 | 6 | 6 | 45 | 0.89 | 49 | 40.4 | 1.11 | 1.11 |
| Nb | 3 | 6 | 6 | 25 | 0.82 | 24 | 32 | 0.84 | 0.84 |
| 80%Nb + 20%Ti | 3 | 7 | 6 | 22 | 0.82 | 22 | 31 | 0.89 | 1.14 |

According to the results shown in Table 3, specimens executed with the fluorides $CaF_2$, LiF and Naf presented, at face and back sides, crack lengths and critical widths significantly lower values than those of the specimen fabricated without powders. This result is ascribed to the higher arc densities obtained with specimens coated with fluorides. Indeed, fluorides migrate to the weld arc leading to a constriction of the arc, according to Howse et al. [24]. The authors suggested that the constriction of the arc will increase the temperature at the anode because of the increase in the current density and of the higher arc voltage. This phenomenon leads to a narrower weld pool [25–27] and with minimal residual stresses [28].

This observation is in good agreement with the results obtained by Goodwin [29] when he compared in his study the effects of TIG, electron beam welding (EBW) and laser welding. He concluded that the crack resistance was progressively higher for TIG, EBW and laser welding. In other hand, the increase in arc voltage increases the peak temperature of the weld metal, reduces its cooling rate and the retained δ-ferrite content 316L stainless steel weld metal is increased. Hence, the δ-ferrite content of the 316L stainless steel weld metal is increased. δ-ferrite is benefit to the hot cracking resistance. The latter is increased owing to the fact that the body-centered cubic ferrite structure has a smaller coefficient of thermal expansion than that of the face centered cubic austenite structure; thereby, shrinkage stresses during the cooling stage are reduced [30].

The Mn coated specimen shows more resistance to solidification cracking (cf. Table 3), probably due to the formation of MnS compounds which are characterized by higher melting points. MnS compounds are also able to avoid the formation of FeS film at the grain boundaries [4]. The presence of Mn in high concentrations can contribute to the formation of low-melting eutectic composition type $Mn-Mn_4P$ with a solidus temperature of 960 °C, as cited by Arata et al. [31]. We notice that the specimen coated with Mn powder exhibits a crack length lesser comparatively to a TIG weld carried out without any additions.

In our study, the specimen welded with the mixed powders (80% Nb + 20% Ti) exhibited, at face and back sides, the lowest crack length at 22 mm, and the crack stopped

at the critical width of 30.8 mm. On the other hand, the specimen coated with pure Nb powder gave, at both sides, crack lengths (25, 24 mm) and critical widths (32, 32 mm) which were lower values than the remaining specimens executed with pure Mn and Ti powders. Hence, specimens executed with Nb powder and those coated with the mixture of powders (80% Nb + 20% Ti) gave the best hot cracking resistance. We notice that the solidification cracking susceptibility decreased with the presence of niobium. This result can be linked to the benefit of Nb, when adding, in the decreasing of the temperature range where both solid and liquid phases coexist [32]. However, many studies [33,34] considered the Niobium as a rich segregant in the form of small particles formed in the interdendritic spaces which refine the dendritic structure, facilitate the filling of the interdendritic spaces and lead to the reduction in hot cracking susceptibility.

The results depicted in Table 4 shows standard deviation (σ), which represents a measure of the amount of variation or dispersion of a set of values. For crack length face side and critical crack width face side measurements, standard deviation is less than 1.2 mm.

Figures 5–7 show that a single crack was formed along the centerline of the bead. Neither transverse cracks nor cracks at the heat-affected zone were detected either during TIG welding with or without adding powders.

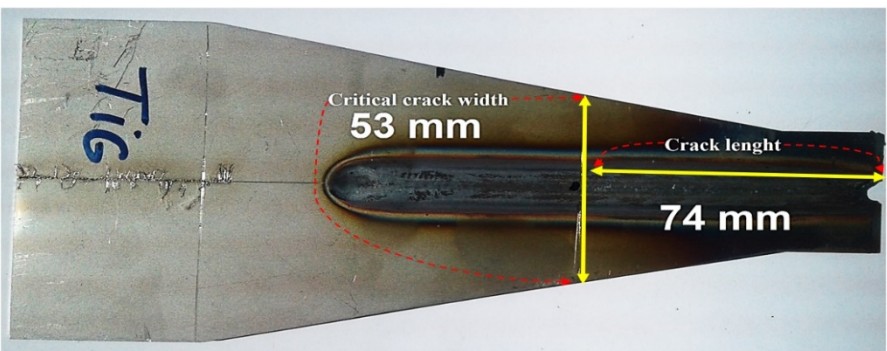

**Figure 5.** TIG-welded specimen without adding powders, face side.

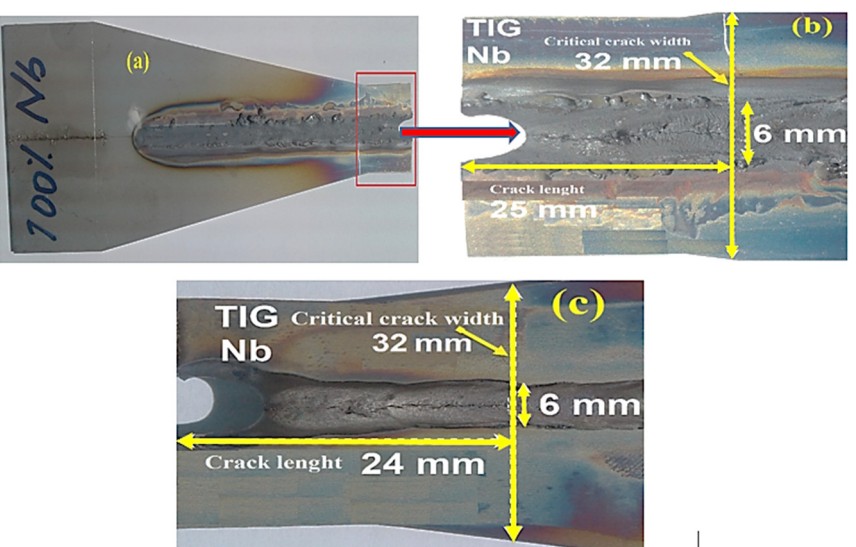

**Figure 6.** Welded specimen coated with Nb powders, face side (**a**), enlarged of marked area (**b**) and back side of enlarged marked area (**c**).

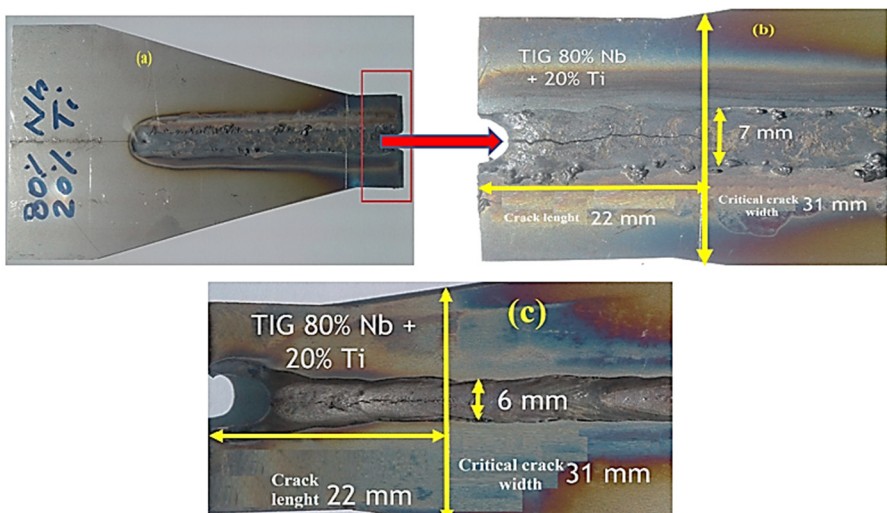

**Figure 7.** Welded specimen coated with the mixture of powders (80% Nb + 20% Ti), face side (**a**), enlarged of marked area (**b**) and back side of enlarged marked area (**c**).

### 3.2. Microstructure Assessment

### 3.2.1. TIG Weld Specimen without Coating Powder

SEM micrographs in Figure 8 shows the dendritic fracture surface of 316L. This is a confirmation that the solidification cracking occurred. The protuberances in the shape of eggs have been formed and surrounded by unfilled cavities. The ratio $Cr_{eq}/Ni_{eq} = 1.55$ for 316L, hence the material solidified in the ferritic-austenite mode. Even if the weld speed could be considered as relatively high, the δ-ferrite proportion remarkably decreased leading to a segregation of P in the liquid metal. Therefore, precipitants as $(Fe, Cr, Ni)_3P$ will be formed and characterized by low melting temperatures (875–1050 °C), as reported by Arata et al. [31]. These precipitants can also widen the brittle range temperature (BTR) at the grain boundary locations. The formation of sulphur-based precipitants is excluded owing to the low sulphur content in the as-received materials. Hence, it is not evident to find sulphides as inclusions or low eutectics as precipitants.

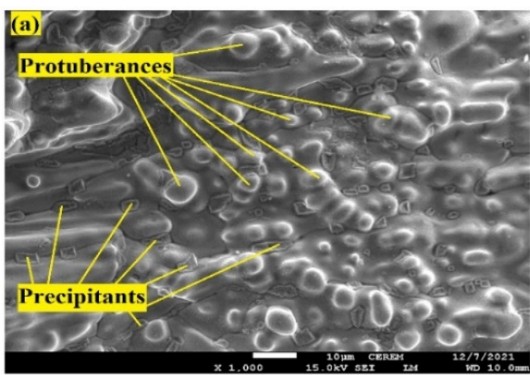
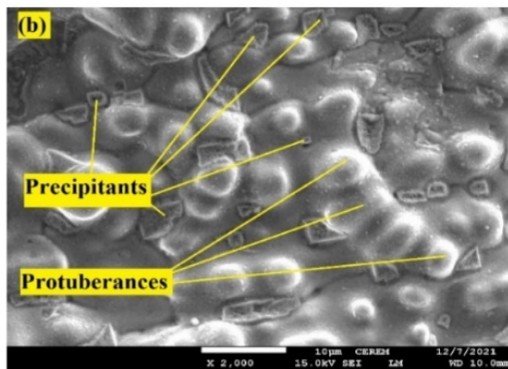

**Figure 8.** SEM micrographs of crack face of TIG weld specimen without coating powder (**a**) (×1000) and (**b**) (×2000).

Figure 9 shows EDS spectra of the lateral solidification cracking face of a specimen welded without coating powder. The spectral result allows us to confirm the absence of precipitants or depletion of Cr and Ni elements. We suspect the formation of chromium carbides as $(Cr, Fe)_{23}C_6$ or $(Cr, Fe)_7C_3$, and/or molybdenum carbides which gives a brittle aspect to the microstructure. Phosphorus (P), which can induce solidification cracks, was observed. The oxygen(O) present in the EDS result is probably aspired from the

environment towards the weld pool during the formation of the crack despite the shield gas protection used at the face side and back side of the weld pool.

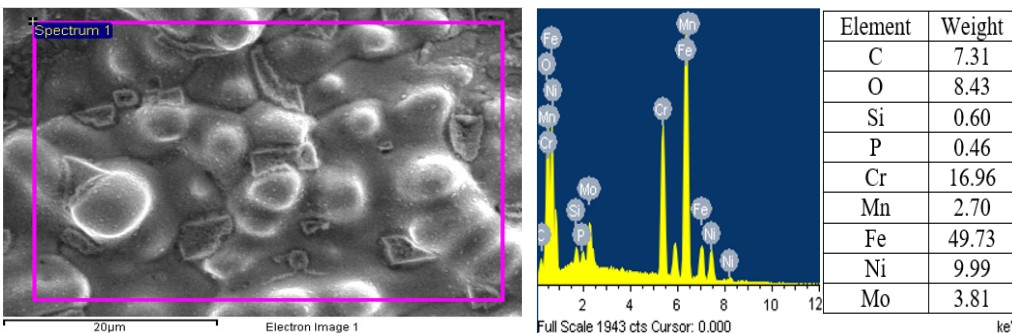

| Element | Weight |
|---------|--------|
| C | 7.31 |
| O | 8.43 |
| Si | 0.60 |
| P | 0.46 |
| Cr | 16.96 |
| Mn | 2.70 |
| Fe | 49.73 |
| Ni | 9.99 |
| Mo | 3.81 |

**Figure 9.** EDS spectral analysis in the lateral solidification cracking face of the specimen welded without coating powder.

### 3.2.2. TIG Weld Specimen Coated with Nb Powder

We noticed in the micrographs of Figure 10 the multiple bumps in the shape of eggs attesting that the crack solidification occurred. The appearance of backfilling is evident in Figure 10b. The metal liquid fills the spaces between protuberances, leading to a decrease in the crack length and, subsequently, the critical width. This phenomenon is the healing of the crack, which is produced by the addition of niobium.

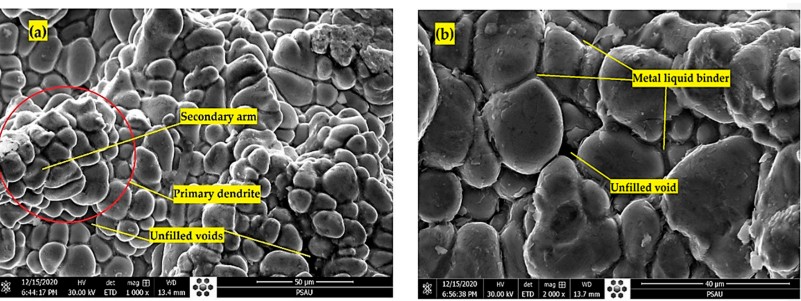

**Figure 10.** SEM micrographs of crack face of TIG weld specimen coated with Nb powder (**a**) (×1000) and (**b**) (×2000).

Figure 11 shows SEM micrograph and EDS spectra of the area containing cracks. The several chemical components such as C, Cr, Fe, Mn, Nb, Ni, and O were detected. We noticed the presence of Nb, C are enhanced and depletion of Cr and Ni. Elements as Nb and C preferentially segregate in the liquid during the solidification stage. Thus, the interdendritic liquid becomes enriched in Nb and C. As the Fe-rich γ dendrites begin to form, less Nb is taken into the solid and, as a result, more is rejected to the liquid. Consequently, the solidification ends at a lower temperature (1223 °C), as mentioned by Dupont [35]. In such cases, BTR is shortened, leading to the healing phenomenon as shown in Figure 12. Thus, the crack length decreases by 2.9 times in comparison to that of the TIG specimens without coating powder. As mentioned above, the suction of oxygen(O) in the weld pool is due to the occurrence of crack.

### 3.2.3. TIG Weld Specimen Coated with the Mixture of Powders (80% Nb + 20% Ti)

The SEM Image in Figure 13a taken at × 500 magnification was performed perpendicularly to the surface crack, showing a top view of primary dendrites and non-filled voids. Figure 13b is taken at × 2000 and shows the backfilling process leading to healing of the cracks. The liquid metal slips between the secondary and ternary dendrites to bind the dendrites and fill the cavities. Owing to the high welding speed and the rapid solidification which follows, the backfilling of the cavities has not come to end, as shown in Figure 13b.

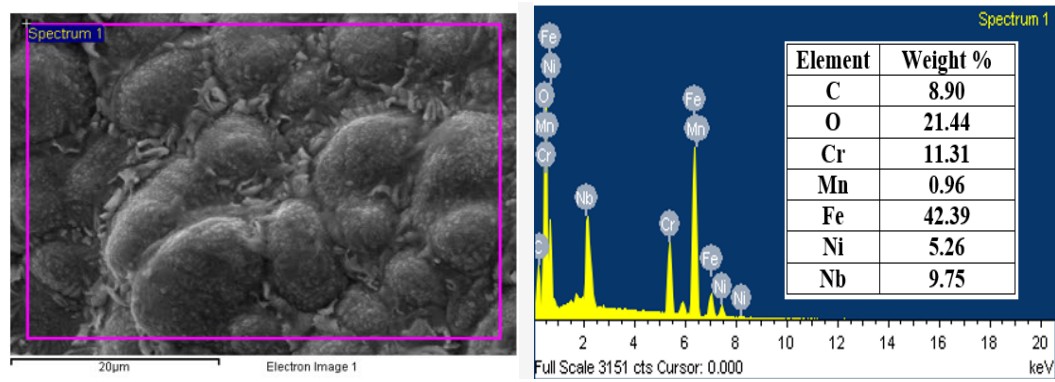

**Figure 11.** EDS spectral analysis in the lateral solidification hot cracking face in TIG weld coated by pure Nb powder.

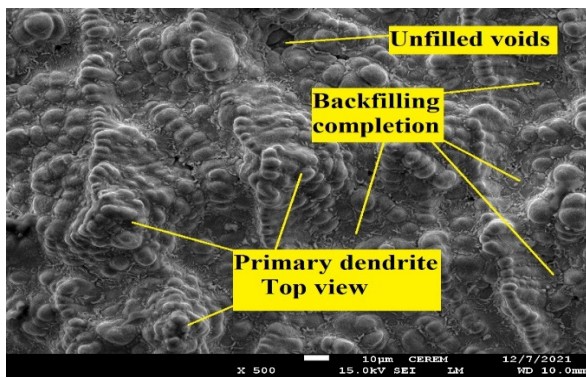

**Figure 12.** SEM micrograph of solidification hot cracking face in TIG weld coated by pure Nb powder (×500).

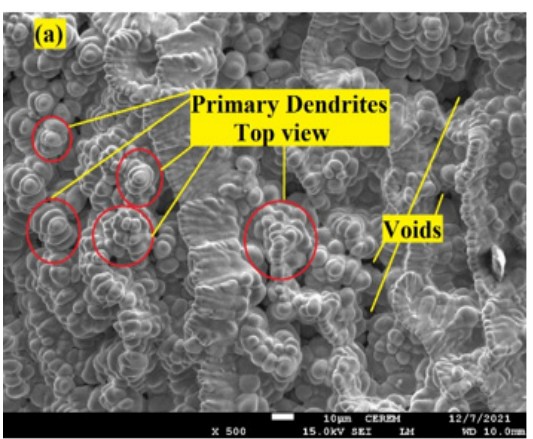 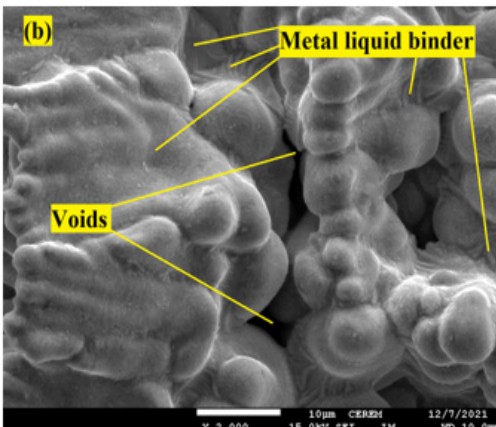

**Figure 13.** Micrograph of crack face of TIG weld specimen coated with the mixture of powders (80% Nb + 20% Ti) (**a**) (×500) and (**b**) (×2000).

Figure 14 shows the measurements of EDS taken to have an overview of elements in presence such as S, P, Ti, Nb, Cr and Ni. The elements Ni and Cr show very little tendency for micro segregation. Ni, Cr and Fe are the main constitutes of the matrix. The carbon (C) and niobium (Nb) components are enriched along the crack surface. Most likely, Nb and C preferentially segregate in the liquid during the solidification stage. Thus, the interdendritic liquid becomes enriched in Nb and C during the primary solidification step as explained previously in the case of the Nb-coated specimen. As the Fe-rich γ dendrites begin to form, less Nb is taken into the solid phase and, as a result, more Nb is rejected to the liquid, as

shown in SEM micrographs presented in Figures 13b and 15. As mentioned above, the presence of oxygen(O) is ascribed to the apparition of the crack.

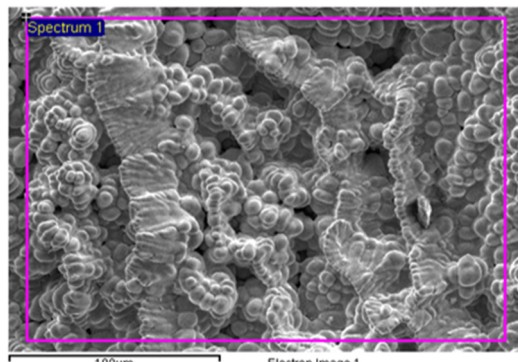 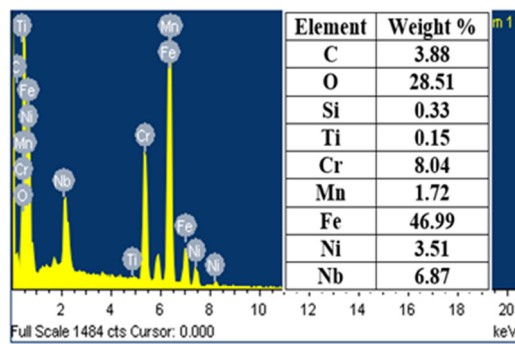

| Element | Weight % |
|---------|----------|
| C | 3.88 |
| O | 28.51 |
| Si | 0.33 |
| Ti | 0.15 |
| Cr | 8.04 |
| Mn | 1.72 |
| Fe | 46.99 |
| Ni | 3.51 |
| Nb | 6.87 |

**Figure 14.** EDS spectral analysis in the lateral solidification hot cracking face in TIG weld coated by the mixture of powders (80% Nb + 20% Ti).

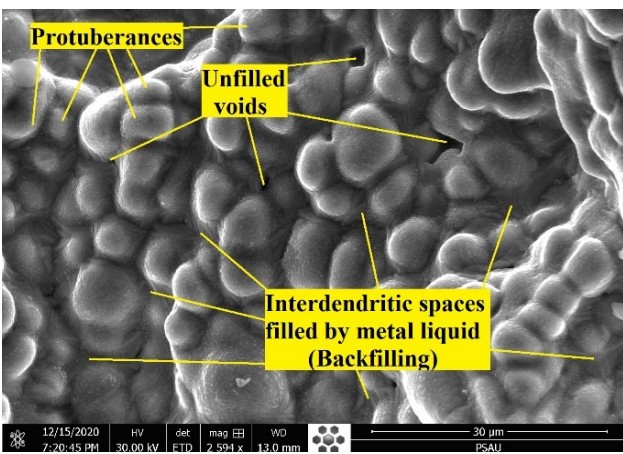

**Figure 15.** SEM micrograph at the solidification cracking surface in TIG weld coated with the mixture of powders (80% Nb + 20% Ti) × 2500.

In addition to SEM and EDS investigations, specimen coated with the mixture of powders 80% Nb and 20% Ti which gave lesser crack length and critical width was studied via X-ray diffraction (XRD). XRD technique revealed the presence of niobium carbide ($Nb_2C$), ($TiO_{0.6}Cr_{0.2}Nb_{0.0202}$) phase and cementite ($Fe_3C$) as depicted in Figure 16. The addition of Nb and Ti causes the formation of niobium carbide and phase composed by niobium-titanium-chromium. It also promotes the replacement of the chromium carbide owing to higher affinity of niobium for carbon in comparison to that chromium. As a result, it was confirmed from the XRD results that the eutectic microstructure of niobium carbide and complex phase were produced in the interdendritic regions.

Owing to a high cooling rate, during the last stage of the solidification, low melting-point constituents segregate between solidifying dendrites and form liquid films over the interface of dendrites. This stage of solidification in which both liquid and solid coexist. Study conducted by Seidai et al. [36] confirms the contribution of Nb in reducing BTR.

The addition of niobium with presence of carbon leads to the decrease in BTR as reported in Vallant's study [33]. Hence, the solidification cracking susceptibility decreases owing to filling of the interdendritic cavities and voids by the precipitants as well as by the continuous supply of liquid metal. This phenomenon leads to the backfilling of cracks which contributes to a decrease in the crack length and the critical width [34].

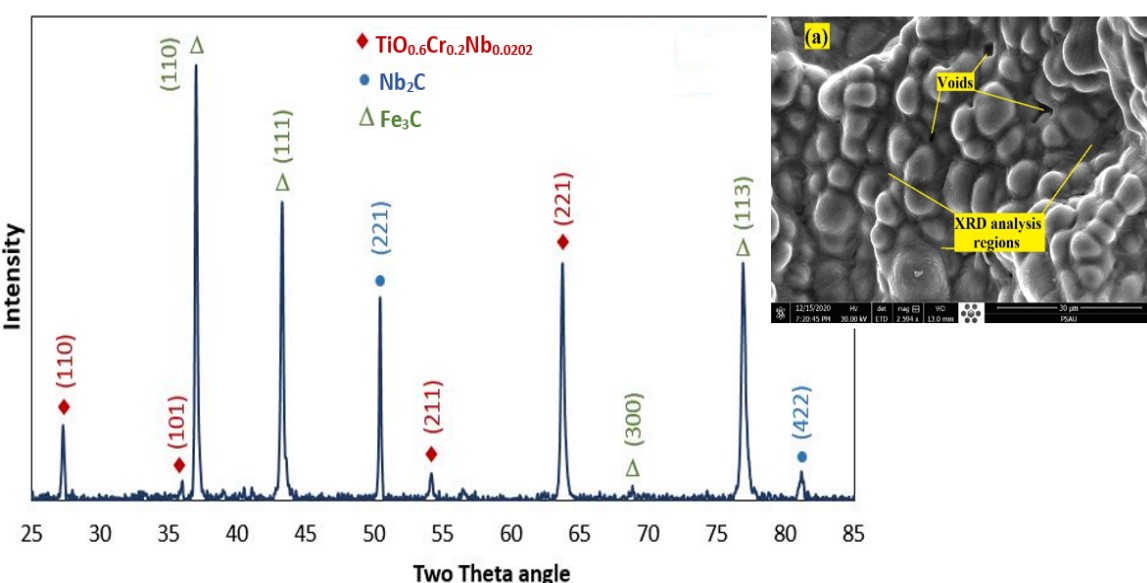

**Figure 16.** X-Ray Diffraction (XRD) pattern in the solidification hot cracking surface in TIG weld coated with the mixture of powders (80% Nb + 20% Ti), (**a**) XRD analysis regions × 2500.

## 4. Conclusions

This work aims to investigate the effects on the hot cracking susceptibility of powders such as $CaF_2$, NaF, LiF, Mn, Ti, Nb and mixed Nb-Ti deposited on 316L stainless steel samples when TIG welding is used and compared to sample without any coating powder. A self-restraint hot cracking test with trapezoidal-shaped samples and 3 mm of thickness was carried out. Based on the obtained results, the following conclusions can be drawn:

(i)     TIG welded specimens without coating powder had a greater tendency to solidification cracking than those in which the powders had been pre-deposited.

(ii)    EDS analyses of specimen without coating powder revealed the presence of phosphorus which can lead to the formation of low melting precipitants such as (Fe, Cr, Ni)$_3$P. The formation of these precipitants increased the gap between solidus and liquidus during weld solidification. The strain developed during the welding operation contributed to increasing the crack length.

(iii)   Fluorides and single powders as Mn and Ti had no beneficial contribution in improving the hot cracking resistance.

(iv)    As Nb content increased in the weld metals by coating the samples, the total crack length decreased by 2.9 times and, subsequently, the critical width decreased by 1.65 times.

(v)     The decrease in solidification cracking susceptibility for specimens coated with Nb or with the mixture of powders (80% Nb + 20% Ti) occurs by adequate backfilling or interdendritic liquid feeding. This compensated both solidification shrinkage and thermal contraction.

(vi)    XRD analyses revealed the presence of Niobium carbide ($Nb_2C$), ($TiO_{0.6}Cr_{0.2}Nb_{0.0202}$) complex phase and cementite ($Fe_3C$) leading to a decreasing in the solidification cracking susceptibility, owing probably to the reduction in the gap between solid and liquid phases during welding operation.

**Author Contributions:** Conceptualization, K.T. and A.C.H.; methodology, K.T. and A.C.H.; software, K.T. and A.C.H.; validation, K.T., A.O. and H.S.A.; formal analysis, K.T.; investigation, K.T., H.S.A., A.O. and A.C.H.; resources, A.I.; data curation, K.T. and A.I.; writing—original draft preparation, K.T.; writing—review and editing, K.T., A.O., A.C.H. and A.I.; visualization, K.T.; supervision, K.T.; All authors have read and agreed to the published version of the manuscript.

**Funding:** This work is funded by the Ministry of Education through the project number (IF-PSAU-2021-01-18492).

**Institutional Review Board Statement:** Not applicable.

**Informed Consent Statement:** Not applicable.

**Data Availability Statement:** Data will be available upon request through the corresponding author. The data used to support the findings of this study are included within the article.

**Acknowledgments:** The authors extend their appreciation to the Deputyship for Research & Innovation, Ministry of Education in Saudi Arabia for funding this research work through the project number (IF-PSAU-2021-01-18492).

**Conflicts of Interest:** The authors declare no conflict of interest.

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
