# Peer review of "Effects of Metal and Fluoride Powders Deposition on Hot-Cracking Susceptibility of 316L Stainless Steel in TIG Welding"

_metals, doi:10.3390/met12071225_

Round 1

Reviewer 1 Report

Occurrence of hot-cracking is a big problem in the welding of a stainless steel. TIG welding is a popular method used in stainless welding. The authors investigated the measures, such as depositing fluoride and metal powders on the surface of a stainless steel, to decrease the tendency of hot-cracking. The obtained result is of importance in practical engineering. It is worth for publication if the following issues are considered.

1, The result shows that some measures, for example, mixed metal powders are effective in preventing hot-cracking. However, this conclusion was obtained from only one welding condition (Table 3). Why this welding condition was used? Moreover, the effective range of welding conditions should be given.  

Author Response

Reviewer 1

On behalf of all the contributing authors, I would like to express our sincere appreciation of your constructive comments concerning our article entitled Effects of Metal and Fluoride Powders Deposition on Hot-cracking Susceptibility of 316L Stainless Steel in TIG Welding. These comments are all valuable and helpful for improving our article.

Comment

The result shows that some measures, for example, mixed metal powders are effective in preventing hot-cracking. However, this conclusion was obtained from only one welding condition (Table 3). Why were this welding conditions used? Moreover, the effective range of welding conditions should be given. 

Response

Preliminary tests were conducted in order to meet the criteria required for validity of crack test which are:

- A continuous total penetration along the line of welding.

- The face width and the root width of the bead must be slightly the same.

- An appearance of the crack.

Once the occurrence of crack is realized with the above-mentioned criteria met, the welding parameters were adopted for all the study. We notice that the starting specimen was that carried out without any deposition of powder. Our target was to decrease the length of the crack with the addition of different available powders. Based on the specimen dimensions, these welding parameters provide quantitative and reproducible results (cracks length, critical cracks width, face width and root width).

your comments are relevant and may be a subject of research where the effect of welding parameters will be studied with the deposit which gives the least crack length. The impact of welding parameters can also be considered through their influence on the temperature gradient.

Reviewer 2 Report

Is better to write the units in the form mg.cm-2, not mg/cm2. Please, check and correct it in all text.

In Table 3 you to correct the unit - 45o

The δ-ferrite or δ ferrite is correct? (rows 238-240, 286, 300 etc.).

Please, to edit the Figure 5, 6 a 7. The describe is not proportional to size of figures. Text have not same size and I think, it is not necessere to write it into yellow filds.

Row 300 - Figure 9., not Figure 9:

Describing of graph lines is not good readable, plase edit it.

Author Response

Reviewer 2

On behalf of all the contributing authors, I would like to express our sincere appreciation of your constructive comments concerning our article entitled Effects of Metal and Fluoride Powders Deposition on Hot-cracking Susceptibility of 316L Stainless Steel in TIG Welding. These comments are all valuable and helpful for improving our article.

Comment 1

Is better to write the units in the form mg.cm-2, not mg/cm2. Please, check and correct it in all text.

Response 1

Corrected as suggested.

Comment 2

In Table 3 you to correct the unit - 45o

Response 2

Corrected as suggested.

Comment 3

The δ-ferrite or δ ferrite is correct? (rows 238-240, 286, 300 etc.).

Response 3

δ ferrite is replaced by δ-ferrite in overall the manuscript.

Comment 4

Please, to edit the Figure 5, 6 a 7. The describe is not proportional to size of figures. Text have not same size and I think, it is not necessary to write it into yellow fields.

Response 4

Figures 5, 6 and 7 re-edited as suggested, unifying the size of text, removing of yellow fields.

Comment 5

Row 300 - Figure 9., not Figure 9:

Response 5

Corrected as suggested

Comment 6

Describing of graph lines is not good readable, please edit it.

Response 6

The graph in Figure 16 replaced.

Reviewer 3 Report

This manuscript reports on Effects of Metal and Fluoride Powders Deposition on Hot-cracking Susceptibility of 316L Stainless Steel in TIG Welding. Authors should improve introduction and discussion parts. I recommend a major revision.

1.     Please check in the entire manuscript that everything is written correctly (subscript should be used for example Creq, instead of Amp or Volts should be A and V (Table 3)).

2.     In introduction literature overview of similar research, using coatings is not sufficiently covered and should be expanded (with other methods). Why was this deposition method chosen and what are the possible advantages of these coatings?

3.     In table 2 the size of used particles and form should be added. Normally, I would also add SEM of each powder.

4.     In Figure 3b it is not possible to see anything. Please add a scale bar.

5.     Even taking into consideration that this paper is about welding, authors should add more details regarding preparation of coatings.

6.     In table 4 there is no standard deviation. What is about repetitiveness of these results?

7.     How did you come to Nb+Ti combination?  Results of Nb coating is practically comparable with Nb+Ti mix.

8.     All SEM Figures should be presented in the same way. In case of EDX there is no sense in representation of chemical composition of entire region. It should be a few points or mapping instead.

9.     Quality of Figure 16 is unacceptable and should be within the format of all other Figures. XRD should be provided as well for Nb and weld without coating for comparison

10.  Entire discussion part is too short. 

Author Response

Reviewer 3

On behalf of all the contributing authors, I would like to express our sincere appreciation of your constructive comments concerning our article entitled Effects of Metal and Fluoride Powders Deposition on Hot-cracking Susceptibility of 316L Stainless Steel in TIG Welding. These comments are all valuable and helpful for improving our article.

Comment 1

Please check in the entire manuscript that everything is written correctly (subscript should be used for example Creq, instead of Amp or Volts should be A and V (Table 3)).

Response 1

Corrected as suggested

Comment 2

 In introduction literature overview of similar research, using coatings is not sufficiently covered and should be expanded (with other methods). Why was this deposition method chosen and what are the possible advantages of these coatings?

Response 2

 Other methods and references are added in introduction .

 To deposit the powders we used a paintbrush. This method is more simple , flexible and cheaper in comparison to spray method or convening the powder through shield gas. details regarding preparation of coatings is added in paragraph (2.2 welding procedure).

Comment 3

In table 2 the size of used particles and form should be added. Normally, I would also add SEM of each powder.

Response 3

The size and form of powders are is added in paragraph (2.2 welding procedure).

Unfortunately we apologize to not be able to do SEM of each powder. SEM machine is under annual maintenance.

Comment 4

In Figure 3b it is not possible to see anything. Please add a scale bar.

Response 4

Figure 3b is improved and scale bar is added as suggested.

Comment 5

Even taking into consideration that this paper is about welding, authors should add more details regarding preparation of coatings.

Response 5

Details about  preparation of coating is added in paragraph titled “welding procedure“ in the manuscript.

Comment 6

In table 4 there is no standard deviation. What is about repetitiveness of these results?

Response 6

For each case (sample without coating and specimens with powder deposit) 3 tests were performed. The values depicted in table 4 represent the average of these three tests. The values were very close attesting  the high confidence level of the conducted tests. The number of tests is added in table 4. The standard deviation is also added for crack length and the critical crack width only.  Both parameters were used to  assess the hot cracking susceptibility. A sentence relevant to standard deviation is added in the manuscript from line 271 to 273 . A sentence mentioning the number of test for each case is added in paragraph (2.2 welding procedure).

Comment 7

How did you come to Nb+Ti combination?  Results of Nb coating is practically comparable with Nb+Ti mix.

Response 7

First we tested Nb as a single deposit powder on the trapezoidal plate, the results was improved in comparison to fluorides or Mn, Ti or specimen without any coating. then we used a mixed powder  80 % Nb + 20 % Ti again the crack length and the critical crack width were slightly improved. This mixed combination was a trial to prospect the effect of this combination with high proportion for Nb since we got a sharp decrease in length crack in comparison with the remaining powders. Matsuda et al [1] found that Ti addition up to 0.05% decreased the hot cracking tendency by increasing the melting point of low-melting phosphide eutectics. Morishige et al. [2] found that niobium decreases hot ductility in the HAZ.

[1] Matsuda, F., Kat lyama, S. and Arata. Y. 1983. Solidification Crack Susceptibility in Weld Metals of Fully Austenitic Stainless Steels (Report IX) - Effect of Titanium on Solidification Crack Resistance. Trans. JWRI 12, 87-92.

[2] Morishige, N., Kuribayashi, M. and Okabashi, H. 1979. Effects of chemical composition of base metal on susceptibility to hot cracking in stainless steel welds. IIW Document IX-1114-79.

Comment 8

All SEM Figures should be presented in the same way. In case of EDS there is no sense in representation of chemical composition of entire region. It should be a few points or mapping instead.

Response 8

All SEM Figures were presented in same way as suggested. About The EDS measurements, Authors are totally agree with your valuable comment in fact the EDS measurements  were taken to have an overview of elements in presence such as P, Ti , Nb. The EDS technique is more qualitative and determines the approximate weight fraction. EDS can either qualitatively determine the elements that should be measured by displaying them, leading the analyst toward which elements (major, minor and trace) should be measured. In other words,  EDS is performed only to prospect the elements available in these regions. However, the sample coated with mixed powder 80%Nb +20 % Ti which gave least crack length;  Authors used XRD tool which is more accurate about chemical composition to analyze in interdendritic region as shown in Figure 16.

Comment 9

 Quality of Figure 16 is unacceptable and should be within the format of all other Figures. XRD should be provided as well for Nb and weld without coating for comparison

Response 9

Quality of Figure 16 is improved. The XRD was taken only for specimen which gave the lowest crack length and critical crack width that why we focus only mixed powders specimen(80% Nb + 20% Ti) and we added the XRD for Nb deposit powder specimen . Unfortunately we are not able to do The XRD analysis for specimen without coating and specimen with Nb coating. The machine is under annual maintenance and the time allowed to submit the revision version is not sufficient, so once again authors apologize for that.

Comment 10

Entire discussion part is too short. 

Response 10

Discussion is improved.

Reviewer 4 Report

This paper studies effects of deposition of different metal powders on the centerline of the trapezoidal specimen on the solidification cracking resistance. I find this paper interesting and scientifically significant since there are no comparative studies in the published literature devoted to add external agents to reduce or avoid solidification cracking.

Authors should write the total number of samples as well as the number of replicates with tested with the same conditions as this has a significant impact on the credibility of the research results.

Reviewer suggests careful reading of text and typing error corrections like (line 3 – remove a dot at the end of the title, In table 3 - type correct unit designations, and replace title range since values are not given in a range).

Author Response

Reviewer 4

On behalf of all the contributing authors, I would like to express our sincere appreciation of your constructive comments concerning our article entitled Effects of Metal and Fluoride Powders Deposition on Hot-cracking Susceptibility of 316L Stainless Steel in TIG Welding. These comments are all valuable and helpful for improving our article.

Comment 1

Authors should write the total number of samples as well as the number of replicates with tested with the same conditions as this has a significant impact on the credibility of the research results.

Response 1

For each case (sample without coating and specimens with powder deposit) 3 tests were performed. The values depicted in table 4 represent the average of these three tests. The values were very close attesting the high confidence level of the conducted tests. The number of tests is added in table 4. The standard deviation is also added for crack length and the critical crack width only.  The crack length and the critical crack width owing to the later parameters are the factors used to assess the hot cracking susceptibility. A sentence mentioning the number of tests for each case is added in paragraph (2.2 welding procedure).

Comment 2

Reviewer suggests careful reading of text and typing error corrections like (line 3 – remove a dot at the end of the title, In table 3 - type correct unit designations, and replace title range since values are not given in a range).

Response 2

The manuscript was reviewed. The word range was replaced by value in the table 3.

Round 2

Reviewer 1 Report

The authors have well revised their papers. I agree to publish in the present form.

Author Response

Dear Reviewer,

We do appreciate your positive feedback and the constructive comments you have provided that improved our manuscript significantly. Again, the authors thanks you very much for reviewing the manuscript stay at your disposal if you have any suggestions.

Reviewer 3 Report

Authors improved the manuscript, but before acceptance XRD graph should be reproduced using Origin or at least Excel in order to improve readability of the curve. The current version is not acceptable. 

Another question is still the combination of Nb and Ti powders used in this work. In my opinion, improvement is not as high as I mentioned before, but the authors added citation where 0.05% of Ti improved resistance against hot cracking, which is significantly less than used in this work. In this case, it should be performed a gradual increase in Ti to ensure optimization of the process.

Best regards,

AVO

Author Response

Dear Reviewer,

We do appreciate your positive feedback and the constructive comments you have provided that improved our manuscript significantly. Again, the authors thanks you very much for reviewing the manuscript stay at your disposal if you have any suggestions.

Comment 1

Authors improved the manuscript, but before acceptance XRD graph should be reproduced using Origin or at least Excel in order to improve readability of the curve. The current version is not acceptable. 

Response 1

.As requested, the XRD graph has been reproduced using Excel in order to improve readability of the curve. Moreover, all peaks in the pattern have been identified according to DIFFRAC.EVA analysis software.as suggested.

Comment 2

Another question is still the combination of Nb and Ti powders used in this work. In my opinion, improvement is not as high as I mentioned before, but the authors added citation where 0.05% of Ti improved resistance against hot cracking, which is significantly less than used in this work. In this case, it should be performed a gradual increase in Ti to ensure optimization of the process.

Response 2

your comment is judicious and relevant to vary the titanium rate gradually to properly measure the impact of the titanium addition on hot cracking. the authors think that this idea can constitute a good topic for research. In this case the mixing method design is more appropriate method to test many combination to optimize the best mixture which gives a least crack length.

Round 3

Reviewer 3 Report

None